# Design, Synthesis, Antibacterial, and Antifungal Evaluation of Phenylthiazole Derivatives Containing a 1,3,4-Thiadiazole Thione Moiety

**DOI:** 10.3390/molecules29020285

**Published:** 2024-01-05

**Authors:** Guoqing Mao, Yao Tian, Jinchao Shi, Changzhou Liao, Weiwei Huang, Yiran Wu, Zhou Wen, Linhua Yu, Xiang Zhu, Junkai Li

**Affiliations:** 1Engineering Research Center of Ecology and Agricultural Use of Wetland, Ministry of Education, Hubei Key Laboratory of Waterlogging Disaster and Agricultural Use of Wetland, College of Agriculture, Yangtze University, Jingzhou 434025, China; maoguoqing1688@163.com (G.M.); yaotien@163.com (Y.T.); shijinchao1996@163.com (J.S.); liaochangzhou58@163.com (C.L.); 18381914696@163.com (W.H.); wu18230085332@163.com (Y.W.); wenzhou1218@163.com (Z.W.); linhuayu531@sina.com (L.Y.); 2Institute of Pesticides, Yangtze University, Jingmi Road 88, Jingzhou 434025, China; 3National Key Laboratory of Green Pesticide, Key Laboratory of Green Pesticide and Agricultural Bioengineering, Ministry of Education, Guizhou University, Guiyang 550025, China

**Keywords:** thiasporine A, phenylthiazole, 1,3,4-thiadiazole, antibacterial activity, antifungal activity, structure-activity relationship

## Abstract

To effectively control the infection of plant pathogens, we designed and synthesized a series of phenylthiazole derivatives containing a 1,3,4-thiadiazole thione moiety and screened for their antibacterial potencies against *Ralstonia solanacearum*, *Xanthomonas oryzae* pv. *oryzae*, as well as their antifungal potencies against *Sclerotinia sclerotiorum*, *Rhizoctonia solani*, *Magnaporthe oryzae* and *Colletotrichum gloeosporioides*. The chemical structures of the target compounds were characterized by ^1^H NMR, ^13^C NMR and HRMS. The bioassay results revealed that all the tested compounds exhibited moderate-to-excellent antibacterial and antifungal activities against six plant pathogens. Especially, compound **5k** possessed the most remarkable antibacterial activity against *R. solanacearum* (EC_50_ = 2.23 μg/mL), which was significantly superior to that of compound **E1** (EC_50_ = 69.87 μg/mL) and the commercial agent Thiodiazole copper (EC_50_ = 52.01 μg/mL). Meanwhile, compound **5b** displayed the most excellent antifungal activity against *S. sclerotiorum* (EC_50_ = 0.51 μg/mL), which was equivalent to that of the commercial fungicide Carbendazim (EC_50_ = 0.57 μg/mL). The preliminary structure-activity relationship (SAR) results suggested that introducing an electron-withdrawing group at the meta-position and ortho-position of the benzene ring could endow the final structure with remarkable antibacterial and antifungal activity, respectively. The current results indicated that these compounds were capable of serving as promising lead compounds.

## 1. Introduction

Plant pathogens possess an extremely infectious ability, resulting in a high incidence of plant mortality, which severely affect agricultural production and significantly reduce crop yield and quality [1,2,3,4]. For instance, *R. solanacearum*, belongs to a common Gram-negative opportunistic pathogen, can infect an array of crop species including rice, ginger, tomato, potato and tobacco [5,6]. *S*. *sclerotiorum*, a necrotrophic phytopathogenic fungus distributed globally, is capable of causing various symptoms, such as stem rot and pod reduction, leading to annual production losses of 10–50% [7,8]. At present, the application of chemical pesticides is the most effective measure to manage bacterial and fungal diseases of crops [9,10]. However, the misuse and overuse of many conventional pesticides for a long time have created ever-rising resistance, resulting in a significant decrease in the control efficacy of commercial agents [11,12]. Thus, it is urgently needed to develop novel and efficient antibacterial and antifungal agents for the control of crop bacterial and fungal diseases.

Structural optimization based on natural products has been an important way to discover novel pesticides, which is of great significance to delay the development of resistance and enhance pharmacodynamic effects [13,14]. Thiasporine A (Figure 1) is a heterocyclic natural product initially isolated from marine-derived *Actinomycetospora chlora* SNC-032 by MacMillan in 2015 [15]. In our previous work, we synthesized several series of thiasporine A derivatives with potent antifungal activity and demonstrated that phenylthiazole was a promising antifungal activity skeleton [16]. Notably, Shi [17] introduced an 1,3,4-oxadiazole thione into phenylthiazole to synthesize compound **E1** (Figure 1), exhibiting the most excellent antifungal activity against *S. sclerotiorum* with an EC50 value of 0.22 μg/mL, which was superior to that of the commercial fungicide Carbendazim (EC_50_ = 0.70 μg/mL). Regrettably, compound **E1** did not perform excellent antibacterial activity, but this result provides valuable guidance for carrying out subsequent molecular design.

1,3,4-thiadiazole is an important class of five-membered heterocycle ring containing nitrogen and sulfur with broad biological activities as antimicrobial [18], anti-inflammatory [19], antibacterial [20], insecticidal [21] and herbicidal [22], widely applied in the fields of pharmaceutical and agricultural chemistry. Especially in agricultural antibacterial agents, considering 1,3,4-thiadiazole as a “privileged” scaffold, large numbers of commercial pesticides have been developed ceaselessly since 1950s, such as Bismerthiazol and Thiodiazole copper (Figure 1) [23,24,25]. Motivated by the above observation, we intended to introduce an antibacterial scaffold 1,3,4-thiadiazole into phenylthiazole active skeleton to discover and synthesize a series of compounds with antibacterial and antifungal potency (Figure 2). All the target compounds were assayed for their antibacterial activities against *R. solanacearum* and *Xoo*, as well as their antifungal activities against *S. sclerotiorum*, *R. solani*, *M. oryzae* and *C. gloeosporioides*, and the preliminary SARs of these compounds were discussed.

## 2. Results and Discussion

### 2.1. Chemistry

The synthetic steps of the target compounds **5a**–**5p** are outlined in Figure 1. In brief, benzonitrile **1a** was provided by professional suppliers and used as the starting material. Intermediate **2a** was harvested via the reaction of benzonitrile **1a**, magnesium chloride hexahydrate and sodium hydrosulfide hydrate in *N*, *N* dimethylformamide [26]. Subsequently, intermediate **3a** was afforded via the reaction of intermediate **2a** and ethyl 3-bromopyruvate in ethanol [27]. The corresponding intermediate **3a** was reacted with hydrazine hydrate in methanol to synthesize intermediate **4a** by a hydrazinolysis reaction [28]. Finally, intermediate **4a** was ring-cyclized with carbon disulfide and potassium hydroxide under 98% sulfuric acid under ice-salt bath conditions to obtain the target compound **5a** [29]. It was of great importance for the last step to keep the temperature at 0 °C. The synthesized target compounds were obtained with yields of 80–90%. The structures of all target compounds were confirmed utilizing ^1^H NMR, ^13^C NMR, and HRMS. All corresponding signals of protons and carbons were recorded in the ^1^H NMR and ^13^C NMR spectra. In the ^1^H NMR spectra of target compounds **5a**–**5p**, the signals around *δ =* 17.37–13.71 ppm suggested the appearance of the N-NH group. The signals *δ =* 191.81–177.23 ppm in the ^13^C NMR indicated the presence of the thione group (C=S). Additionally, the OCF_3_ was observed as a quartet with a large coupling constant. More detailed information is shown in the Appendix A.

### 2.2. Antibacterial Activity

The preliminary antibacterial activities of the target compounds **5a**–**5p** against *R. solanacearum* and *Xoo* were evaluated at concentrations of 200 and 100 μg/mL via the turbidimeter test. The results in Table 1 suggested that most of the target compounds exhibited moderate to remarkable antibacterial activities against *R. solanacearum* and *Xoo*. Among them, compounds **5b**, **5h**, **5i** and **5k** possessed excellent antibacterial activities against *R. solanacearum* at 100 μg/mL with inhibition rates of 92.00%, 93.81%, 94.00% and 100%, respectively, which were superior to those of compound **E1** (79.77%) and the commercial agent Thiodiazole copper (70.22%). Meanwhile, compound **5k** also performed good activity against *Xoo* at 100 μg/mL with an inhibition rate of 72.63%, which was higher than that of compound **E1** (53.96%), but it had a certain gap compared with Thiodiazole copper (94.61%).

Based on the preliminary screening results, the EC_50_ values of compounds **5b**, **5h**, **5i** and **5k** against *R. solanacearum* were further tested. The results were statistically analyzed and shown in Table 2. Satisfactorily, the EC_50_ values of all tested compounds ranged from 2.23 to 40.33 μg/mL. Notably, compounds **5h**, **5i** and **5k** possessed excellent antibacterial activities against *R. solanacearum*, with EC_50_ values of 6.66, 7.20 and 2.23 μg/mL, respectively, which were lower than those of compound **E1** (EC_50_ = 69.87 μg/mL) and the commercial agent Thiodiazole copper (EC_50_ = 52.01 μg/mL). The EC_50_ value of compound **5b** was relatively large, but it was still lower than that of compound **E1** and the Thiodiazole copper.

### 2.3. Antifungal Activity

The antifungal activities of the target compounds **5a**–**5p** against four phytopathogenic fungi (*S. sclerotiorum, R. solani*, *M. oryzae* and *C. gloeosporioides*) were determined at 50 μg/mL utilizing the mycelial growth rate method. The results of antifungal activities were outlined in Table 3 and indicated that most of the target compounds displayed moderate to excellent antifungal activities against each of the test fungi. Particularly, compounds **5b**, **5h**, **5i** and **5p** possessed remarkable antifungal activities against *S. sclerotiorum* with the inhibition rates of 90.48%, 82.14%, 83.63% and 80.06%, respectively, which surpassed that of the commercial agent Thifluzamide (72.92%). Meanwhile, compound **5b** also exhibited good activity against *R. solani*, *M. oryzae* and *C. gloeosporioides*, with the inhibition rates of 72.32%, 55.36% and 52.98%, respectively. The inhibition rates of compounds **5b, 5e**, **5h**, **5i** and **5j** against *R. solani* were more than 50%. Furthermore, selected compounds with activity higher than 80% were further tested for EC_50_ values to evaluate their excellent antifungal activities more accurately. The results in Table 4 indicated that all of them showed more potent antifungal activities compared with Thifluzamide (EC_50_ = 27.24 μg/mL). Especially, the EC_50_ value of compound **5b** against *S. sclerotiorum* was 0.51 μg/mL, which was equivalent to that of the commercial fungicide Carbendazim (EC_50_ = 0.57 μg/mL). From the pictures in Figure 3, it was displayed the antifungal effects of compound **5b** and Carbendazim against *S. sclerotiorum* at different concentrations. It can be intuitively observed that the antifungal activity of compound **5b** against *S. sclerotiorum* was equivalent to Carbendazim at the same concentration.

### 2.4. Preliminary Analysis of Structure-Activity Relationship (SAR)

The preliminary SAR results were deduced from the inhibitory activity data of the antibacterial and antifungal activities shown in Table 1, Table 2, Table 3 and Table 4. The results indicated that the type and position of substituents on the benzene ring had a great impact on antibacterial and antifungal activities. Briefly, introducing an electron-withdrawing group at the meta-position can endow the final structure with more potent antibacterial activity. For example, the antibacterial activity of compound **5k** (R^1^ = 3-OCF_3_) was superior to that of compound **5i** (R^1^ = 3-CH_3_). In addition, introducing the same substituents at different positions of the benzene ring, the meta-position was of great benefit for improving antibacterial activity. The inhibition rates of compounds **5k** (R_1_ = 3-OCF_3_), **5g** (R^1^ = 2-OCF_3_), and **5o** (R^1^ = 4-OCF_3_) against *R. solanacearum* at 100 μg/mL were 100%, 48.26% and 35.11%, respectively. Similarly, introducing an electron-withdrawing group could contribute to promoting antifungal activity, but the same substituent at the ortho-position of the benzene ring took a stronger advantage in improving activity. For instance, the inhibition rates of compounds **5b** (R^1^ = 2-F), **5h** (R^1^ = 3-F) and **5l** (R^1^ = 4-F) against *S. sclerotiorum* were 90.48%, 82.14% and 32.14%, respectively. Moreover, by introducing the ortho-fluoro group of the benzene ring, the antifungal activity of compound **5b** (R^1^ = 2-F) was more potent than that of compound **5c** (R^1^ = 2-Cl) and compound **5d** (R^1^ = 2-Br).

Fortunately, some of the target compounds possessed potent antibacterial and antifungal potencies against *R. solanacearum* and *S. sclerotiorum*, respectively, and the SAR results were summarized. In this study, we introduced an antibacterial active scaffold into the phenylthiazole active skeleton, hoping to further improve antibacterial activity while retaining antifungal activity as far as possible. Compounds **5k** and **5b** exhibited excellent activities against *R. solanacearum* and *S. sclerotiorum* with EC_50_ values of 2.23 and 0.51 μg/mL, respectively. From the results, the purpose of the design has been preliminarily achieved, and it heralds that these compounds have the potential to serve as lead compounds. Regrettably, it is not the same compound that simultaneously possesses optimal antibacterial and antifungal activity. On the other hand, this study lacks further determination of bioactivity as well as the exploration of mechanisms. We will further optimize the structure to enhance activity and explore the mechanism of action in future work.

## 3. Materials and Methods

### 3.1. Materials and Instruments

A commercial bactericide of Thiodiazole copper, commercial fungicides Thifluzamide and Carbendazim were supplied by the College of Agriculture, Yangtze University. All reagents and solvents used in the experiment were provided by professional suppliers and utilized without further purification unless otherwise indicated. ^1^H NMR and ^13^C NMR of all target compounds were performed on a AVANCE DPX 400 spectrometer (Bruker Co., Ltd., Fällanden, Switzerland) using tetramethylsilane (TMS) and dimethyl sulfoxide-*d*_6_ (DMSO-*d*_6_) as an internal standard and solvent, respectively (2.50 ppm for ^1^H and 39.52 ppm for ^13^C). High-resolution mass spectrometry (HRMS) data were acquired from Thermo Scientific Q Exactive (Thermo Fisher Scientific, Bremen, Germany). All reactions were indicated via thin-layer chromatography (TLC) utilizing silica gel 60 GF254 (Qingdao Hai Yang Chemical Co., Ltd., Qingdao, China). The melting point of the target compounds were measured by WRR melting point apparatus, provided by Shanghai Precision Scientific Instrument Co., Ltd., Shanghai, China.

### 3.2. Bacterial and Fungal Strains

Phytopathogenic bacterial and fungi, including *Ralstonia solanacearum* (*R. solanacearum*), *Xanthomonas oryzae* pv. *oryzae* (*Xoo*), *Sclerotinia sclerotiorum* (*S. sclerotiorum*), *Rhizoctonia solani* (*R. solani*), *Magnaporthe oryzae* (*M. oryzae*) and *Colletotrichum gloeosporioides* (*C. gloeosporioides*) were provided by the Agricultural College Plant Pathology Laboratory of Yangtze University.

### 3.3. Chemical Synthesis

#### 3.3.1. General Procedure for the Preparation of Intermediate **2a**

In a 250 mL flask, the compound **1a** (5.0 g, 42.3 mmol) and MgCl_2_·6H_2_O (10.9 g, 53.4 mmol) were dissolved in *N*, *N* dimethylformamide (DMF, 20 mL), and stirred at room temperature for 15 min. Then, NaHS·H_2_O (5.9 g, 106.7 mmol) was added, and the reaction solution was maintained at room temperature for 16 h. After the reaction was completed (indicated by TLC), the solution was extracted 3–4 times directly with ethyl acetate and saturated salt water. Finally, the organic layers were dried with anhydrous sodium sulfate, filtered, and concentrated to acquire crude intermediate **2a** [26].

#### 3.3.2. General Procedure for the Preparation of Intermediate **3a**

In a 250 mL flask, the crude intermediate **2a** (4.0 g, 29.2 mmol) was dissolved completely using ethanol (30 mL). Immediately, 80% ethyl 3-bromopyruvate (5.7 g, 29.2 mmol) was added. The solution was stirred and refluxed for 4–6 h at 70 °C until the reaction was completed (indicated by TLC). Then, the pure intermediate **3a** was yielded via silica gel column chromatography (petroleum ether/EtOAc = 10:1, *v*/*v*) [27].

#### 3.3.3. General Procedure for the Preparation of Intermediate **4a**

In a 250 mL flask, the intermediate **3a** (3.0 g, 12.9 mmol) was dissolved in methanol (30mL), 80% hydrazine hydrate (3.9 g, 77.4 mmol) was slowly added. The mixture was reacted at room temperature for 4 h. After completion of the reaction (indicated by TLC), the ice water was added to the reaction solution until a white solid was precipitated out. Eventually, the crude product was further washed several times with water to afford intermediate **4a** [28].

#### 3.3.4. General Procedure for the Preparation of Target Compound **5a**

Under the condition of an ice salt bath, 98% sulfuric acid (15 mL) was added to a 250 mL flask. When the temperature dropped below 0 °C, intermediate **4a** (2.3 g, 7.2 mmol), potassium hydroxide (0.77 g 15.4 mmol), and carbon disulfide (1.95 g, 25.6 mmol) were added successively. The reaction was stirred for 2 h at 0 °C. After the reaction was completed (indicated by TLC), the solution was poured into a large amount of ice water to precipitate a white solid. This was the target product that we wanted. Subsequently, the target compound **5a** was purified by recrystallizing with methanol and water [29]. The chemical structures of the target compounds were accurately confirmed using ^1^H NMR, ^13^C NMR, and HRMS. The yields, physical properties, melting points and spectral data of target compounds **5a**–**5p** are given below.

**5a.** 83.2% yield; White solid. m. p. >250 °C. ^1^H NMR (400 MHz, DMSO-*d*_6_) *δ*: 14.45 (s, 1H), 8.43 (s, 1H), 8.02–7.87 (m, 2H), 7.62–7.44 (m, 3H). ^13^C NMR (101 MHz, DMSO-*d*_6_) *δ*: 188.38, 169.10, 155.37, 144.85, 132.30, 131.53, 129.85 (2C), 126.86 (2C), 120.54. HRMS (ESI): calcd for C_11_H_7_N_3_S_3_ [M + H]^+^: 277.9875, found, 277.9875.

**5b.** 84.7% yield; White solid. m. p. 238.5–239.5 °C. ^1^H NMR (400 MHz, DMSO-*d*_6_) *δ*: 17.37 (s, 1H), 11.20 (s, 1H), 10.77 (t, *J* = 7.8 Hz, 1H), 10.12 (q, *J* = 7.0 Hz, 1H), 10.03–9.86 (m, 2H). ^13^C NMR (101 MHz, DMSO-*d*_6_) *δ*: 177.23, 161.46 (d, *J* = 5.0 Hz), 159.46 (d, *J* = 250.8 Hz), 156.44, 138.05, 132.81 (d, *J* = 8.8 Hz), 128.46, 125.51 (d, *J* = 9.1 Hz), 125.41 (d, *J* = 3.2 Hz), 119.59 (d, *J* = 11.4 Hz), 116.50 (d, *J* = 21.3 Hz). HRMS (ESI): calcd for C_11_H_6_FN_3_S_3_ [M + H] ^+^: 295.9781, found, 295.9781.

**5c.** 80.1% yield; White solid. m. p. 234.6–234.9 °C. ^1^H NMR (400 MHz, DMSO-*d*_6_) *δ*: 14.70 (s, 1H), 8.58 (s, 1H), 8.18 (dd, *J* = 6.0, 3.6 Hz, 1H), 7.74–7.63 (m, 1H), 7.55 (p, *J* = 6.4 Hz, 2H). ^13^C NMR (101 MHz, DMSO-*d*_6_) *δ*: 189.48, 164.30, 160.39, 143.87, 132.38, 131.40, 131.32, 131.19, 130.51, 128.44, 122.03. HRMS (ESI): calcd for C_11_H_6_ClN_3_S_3_ [M + H]^+^: 311.9485; found, 311.9487.

**5d.** 84.3% yield; Yellow solid. m. p. 222.7–223.7 °C. ^1^H NMR (400 MHz, DMSO-*d*_6_) *δ*: 14.24 (s, 1H), 8.56 (d, *J* = 2.5 Hz, 1H), 8.03 (d, *J* = 7.8 Hz, 1H), 7.82 (d, *J* = 7.9 Hz, 1H), 7.55 (t, *J* = 7.5 Hz, 1H), 7.45 (t, *J* = 7.6 Hz, 1H). ^13^C NMR (101 MHz, DMSO-*d*_6_) *δ*: 188.00, 165.95, 155.55, 143.98, 134.61, 132.66, 132.44, 131.95, 128.77, 121.92, 121.35. HRMS (ESI): calcd for C_11_H_6_BrN_3_S_3_ [M + H]^+^: 355.8980, found, 355.8981.

**5e.** 83.2% yield; White solid. m. p. 229.3–230.1 °C. ^1^H NMR (400 MHz, DMSO-*d*_6_) *δ*: 14.78 (s, 1H), 8.51 (s, 1H), 7.78 (d, *J* = 7.2 Hz, 1H), 7.52–7.29 (m, 3H), 3.32 (s, 3H). ^13^C NMR (101 MHz, DMSO-*d*_6_) *δ*: ^13^C NMR (101 MHz, DMSO-*d_6_*) *δ*: 188.36, 168.85, 155.74, 144.27, 136.62, 132.26, 131.49, 130.77, 130.10, 127.05, 120.66, 21.70. HRMS (ESI): calcd for C_12_H_9_N_3_S_3_ [M + H]^+^: 292.0031, found, 292.0032.

**5f.** 82.7% yield; White solid. m. p.235.6–236.7C. ^1^H NMR (400 MHz, DMSO-*d*_6_) *δ*: 13.71 (s, 1H), 8.39 (s, 1H), 8.23 (d, *J* = 7.8 Hz, 1H), 7.50 (t, *J* = 7.8 Hz, 1H), 7.25 (d, *J* = 8.4 Hz, 1H), 7.13 (t, *J* = 7.6 Hz, 1H), 4.02 (s, 3H). ^13^C NMR (101 MHz, DMSO-*d*_6_) *δ*: 188.07, 163.06, 156.73, 147.66, 143.19, 132.41, 127.97, 121.52, 120.84, 120.65, 112.78, 56.47. HRMS (ESI): calcd for C_12_H_9_N_3_OS_3_ [M + H]^+^: 307.9981, found, 307.9983.

**5g.** 85.6% yield; White solid. m. p. 230.2–230.7 °C. ^1^H NMR (400 MHz, DMSO-*d*_6_) *δ*: 14.83 (s, 1H), 8.60 (s, 1H), 8.30 (dd, *J* = 8.1, 1.7 Hz, 1H), 7.73–7.67 (m, 1H), 7.65–7.59 (m, 2H). ^13^C NMR (101 MHz, DMSO-*d*_6_) *δ*: 187.88, 161.40, 154.78, 145.14, 143.65, 132.39, 129.62, 128.13, 124.58, 121.56, 121.12, 120.01 (q, *J* = 259.1 Hz). HRMS (ESI): calcd for C_12_H_6_F_3_N_3_OS_3_ [M + H]^+^: 361.9698, found, 361.9697.

**5h.** 84.5% yield; White solid. m. p. 219.1–220.0 °C. ^1^H NMR (400 MHz, DMSO-*d*_6_) *δ*: 14.74 (s, 1H), 8.47 (s, 1H), 7.78 (dd, *J* = 17.5, 8.7 Hz, 2H), 7.68–7.52 (m, 1H), 7.39 (t, *J* = 8.3 Hz, 1H). ^13^C NMR (101 MHz, DMSO-*d*_6_) *δ*: 187.90, 167.02 (d, *J* = 3.1 Hz), 162.41 (d, *J* = 245.1 Hz), 154.72, 144.39, 133.88 (d, *J* = 8.0 Hz), 131.60 (d, *J* = 8.5 Hz), 122.66 (d, *J* = 2.6 Hz), 120.83, 117.82 (d, *J* = 21.1 Hz), 112.89 (d, *J* = 23.6 Hz). HRMS (ESI): calcd for C_11_H_6_FN_3_S_3_ [M + H]^+^: 295.9781, found, 295.9782.

**5i.** 84.8% yield; White solid. m. p. 224.9–225.4 °C. ^1^H NMR (400 MHz, DMSO-*d_6_*) *δ*: 14.76 (s, 1H), 8.39 (s, 1H), 7.73 (d, *J* = 8.4 Hz, 2H), 7.45–7.28 (m, 2H), 2.38 (s, 3H). ^13^C NMR (101 MHz, DMSO-*d*_6_) *δ*: 188.33, 169.19, 155.41, 144.76, 139.28, 132.22, 132.20, 129.72, 127.16, 124.06, 120.39, 21.33. HRMS (ESI): calcd for C_12_H_9_N_3_S_3_ [M + H]^+^: 292.0032, found, 292.0031.

**5j.** 83.1% yield; White solid. m. p. 228.9–230.1 °C. ^1^H NMR (400 MHz, DMSO-*d*_6_) *δ*: 14.57 (s, 1H), 8.45 (s, 1H), 7.63–7.38 (m, 3H), 7.12 (d, *J* = 6.5 Hz, 1H), 3.85 (s, 3H). ^13^C NMR (101 MHz, DMSO-*d*_6_) *δ*: 188.40, 168.85, 160.23, 153.27, 144.78, 133.53, 131.11, 120.77, 119.32, 117.21, 111.83, 55.86. HRMS (ESI): calcd for C_12_H_9_N_3_OS_3_ [M + H]^+^: 307.9981, found, 307.9981.

**5k.** 82.6% yield; White solid. m. p. 228.6–228.9 °C. ^1^H NMR (400 MHz, DMSO-*d*_6_) *δ*: 14.80 (s, 1H), 8.51 (s, 1H), 8.00 (d, *J* = 8.3 Hz, 1H), 7.90 (s, 1H), 7.68 (t, *J* = 8.0 Hz, 1H), 7.59–7.52 (m, 1H). ^13^C NMR (101 MHz, DMSO-*d*_6_) *δ*: 187.91, 166.63, 154.65, 148.88 (d, *J* = 2.0 Hz), 144.47, 133.85, 131.66, 125.64, 123.21, 121.13, 120.02 (q, *J* = 256.9 Hz), 118.43. HRMS (ESI): calcd for C_12_H_6_F_3_N_3_OS_3_ [M + H]^+^: 361.9698, found, 361.9698.

**5l.** 81.8% yield; White solid. m. p. >250 °C. ^1^H NMR (400 MHz, DMSO-*d*_6_) *δ*: 14.75 (s, 1H), 8.39 (s, 1H), 8.09–7.82 (m, 2H), 7.34 (t, *J* = 8.8 Hz, 2H). ^13^C NMR (101 MHz, DMSO-*d*_6_) *δ*: 187.87, 167.41, 163.62 (d, *J* = 249.3 Hz), 154.83, 144.30, 128.79 (d, *J* = 8.8 Hz), 128.48 (2C) (d, *J* = 3.0 Hz), 120.12, 116.43 (2C) (d, *J* = 22.3 Hz). HRMS (ESI): calcd for C_11_H_6_FN_3_S_3_ [M + H]^+^: 295.9781, found, 295.9781.

**5m.** 85.2% yield; Yellow solid. m. p.247.7–248.6 °C. ^1^H NMR (400 MHz, DMSO-*d*_6_) *δ*: 14.76 (s, 1H), 8.33 (s, 1H), 7.80 (d, *J* = 8.1 Hz, 2H), 7.30 (d, *J* = 8.0 Hz, 2H), 2.34 (s, 3H). ^13^C NMR (101 MHz, DMSO-*d*_6_) *δ*: 188.30, 169.18, 155.46, 144.65, 141.54, 130.34 (2C), 129.65, 126.73 (2C), 119.96, 21.47. HRMS (ESI): calcd for C_12_H_9_N_3_S_3_ [M + H]^+^: 292.0032, found, 292.0033.

**5n.** 82.1% yield; Yellow solid. m. p. 219.8–221.5 °C. ^1^H NMR (400 MHz, DMSO-*d*_6_) *δ*: 14.76 (s, 1H), 8.41 (d, *J* = 57.9 Hz, 1H), 7.95 (d, *J* = 8.8 Hz, 1H), 7.90 (d, *J* = 8.8 Hz, 1H), 7.11–7.06 (m, 2H), 3.83 (s, 3H). ^13^C NMR (101 MHz, DMSO-*d*_6_) *δ*: 188.19, 168.97, 163.10, 161.89, 154.39, 145.69, 144.58, 125.08 (2C), 119.55 (2C), 55.93. HRMS (ESI): calcd for C_12_H_9_N_3_OS_3_ [M + H]^+^: 307.9981, found, 307.9981.

**5o.** 83.5% yield; White solid. m. p. 246.8–247.6 °C. ^1^H NMR (400 MHz, DMSO-*d*_6_) *δ*: 14.75 (s, 1H), 8.42 (s, 1H), 8.03 (d, *J* = 8.7 Hz, 2H), 7.47 (d, *J* = 8.3 Hz, 2H). ^13^C NMR (101 MHz, DMSO-*d*_6_) *δ*: 187.88, 166.88, 154.69, 149.90, 144.48, 130.84, 128.43 (2C), 121.62 (2C), 120.62, 119.94 (q, *J* = 257.4 Hz). HRMS (ESI): calcd for C_12_H_6_F_3_N_3_OS_3_ [M + H]^+^: 361.9698, found, 361.9698.

**5p.** 86.4% yield; Gray solid. m. p. >250 °C. ^1^H NMR (400 MHz, DMSO-*d*_6_) *δ*: 14.85 (s, 1H), 8.33 (s, 1H), 7.59 (dd, *J* = 8.0, 1.3 Hz, 1H), 7.19 (t, *J* = 7.7 Hz, 1H), 6.87 (d, *J* = 8.3 Hz, 2H), 6.63 (t, *J* = 7.5 Hz, 2H). ^13^C NMR (101 MHz, DMSO-*d*_6_) *δ*: 188.08, 170.58, 155.20, 146.97, 143.53, 132.09, 129.52, 118.18, 117.23, 116.39, 113.28. HRMS (ESI): calcd for C_11_H_8_N_4_S_3_ [M + H]^+^: 292.9984, found, 292.9985.

### 3.4. Antibacterial Activity Test In Vitro

The preliminary antibacterial activities of the target compounds **5a**–**5p** against *R. solanacearum* and *Xoo* were assayed at 200 and 100 μg/mL by the turbidimeter test [30]. Nutrient broth (NB) mediums containing 0.5% Dimethylsulfoxide (DMSO) and 0.1% Tween 80 served as blank control, whereas compound **E1** and commercial agent Thiodiazole copper were used as positive controls. Firstly, each tested compound (20 mg) was dissolved in a sterilized tube with a mixture solution of DMSO (200 μL) and 0.1% Tween 80 (10 μL). The solution was diluted with sterile water and added to NB mediums to obtain the final concentrations (200, 100 μg/mL). Subsequently, NB mediums containing *R. solanacearum* and *Xoo* were added to the sterilized test tubes and then incubated on a shaker for 24–72 h at 30 °C and 180 rpm. Each treatment was tested in triplicate. The optical density at 595 nm was measured when the *R. solanacearum* or *Xoo* was in the untreated culture in the logarithmic phase. The inhibition rates were calculated via the following Formula (1).
Inhibition rate (%) = (C − T)/C × 100(1)
where C is the corrected turbidity value of the untreated NB medium, and T is the corrected turbidity value of the treated NB medium. The growth inhibition rates and the standard errors were calculated by Microsoft Excel 2016 (Version 16.0.17029.20028) software.

### 3.5. Antifungal Activity Test In Vitro

The antifungal activities of the target compounds **5a**–**5p** against four plant pathogenic fungi (*S. sclerotiorum*, *R. solani*, *M. oryzae* and *C. gloeosporioides*) were evaluated at 50 μg/mL utilizing the mycelial growth inhibitory rate method [31]. PDA mediums containing 0.5% DMSO and 0.1% Tween 80 were employed as a blank control, and the commercial fungicides Thifluzamide and Carbendazim served as positive controls. Firstly, each tested compound (15 mg) was dissolved in DMSO (200 μL) containing 0.1% Tween 80 and diluted with sterile water. Secondly, the prepared solution was added to sterile molten PDA mediums to obtain a final tested concentration (50 μg/mL). The PDA mediums containing the corresponding medicinal solution were poured into sterile Petri plates per plate (15 mL). Thirdly, the 7-mm-diameter mycelial discs of fungi were inoculated in the center of the PDA Petri plates after cooling. The inoculated medium was cultured at 26 ± 2 °C. Three replicates were conducted for each treatment. The mycelium diameters (mm) of each treatment were accurately measured when the blank control reached two-thirds of the Petri plates. The inhibition rates of the tested compounds were calculated by the following Formula (2).
Inhibition rate (%) = [(C_1_ − T_1_)/(C_1_ − 7 mm)] × 100(2)
where C_1_ is the average colony growth diameter of the blank control, T_1_ is the average colony growth diameter of treatment, and 7 mm is the diameter of mycelial discs. The growth inhibition rates and the standard errors were calculated by Microsoft Excel 2016 (Version 16.0.17029.20028) software.

Selected target compounds with remarkable antibacterial and antifungal potencies were further assayed for their median effective concentration (EC_50_) values. Based on the above-mentioned methods and the screening results, PDA mediums containing 25, 15, 10, 5.0, 2.5, 1.2, 0.8, 0.6, 0.5, 0.4 μg/mL of the tested compounds and NB mediums containing 100, 80, 60, 50, 30, 20, 10, 8, 5, 2.5, 1 μg/mL of the tested compounds were prepared, respectively, and their antibacterial and antifungal activities were indicated via assaying the inhibition rates against *R. solanacearum* and *S. sclerotiorum*. The log dose–response curves allowed for the determination of the EC_50_ values by utilizing Data Processing System (DPS) (Version 20.05) software.

## 4. Conclusions

In summary, 16 phenylthiazole derivatives containing a 1,3,4-thiadiazole thione moiety were designed and synthesized based on the phenylthiazole active skeleton. Their chemical structures were characterized by ^1^H NMR, ^13^C NMR and HRMS. The bioassay results indicated that most of the target compounds displayed moderate to remarkable antibacterial and antifungal activities. In particular, compound **5k** possessed the most excellent antibacterial activities against *R. solanacearum* with an EC_50_ value of 2.23 μg/mL, which was superior to that of compound **E1** (EC_50_ = 69.87 μg/mL) and the commercial bactericide Thiodiazole copper (EC_50_ = 52.01 μg/mL). Meanwhile, compound **5b** exhibited the most potent antifungal activity against *S. sclerotiorum* with an EC_50_ value of 0.51 μg/mL, which was equivalent to that of the commercial fungicide Carbendazim (EC_50_ = 0.57 μg/mL). The preliminary SAR results indicated that introducing an electron-withdrawing group at the meta-position and ortho-position of the benzene ring was more favorable for promoting antibacterial and antifungal activity, respectively. The current results provide a new strategy for developing effective antibacterial and antifungal drugs for managing bacterial and fungal diseases in crops.

## Data Availability

Data are contained within the article and Appendix A.

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
