# Peer review of "Design, Synthesis, Antibacterial, and Antifungal Evaluation of Phenylthiazole Derivatives Containing a 1,3,4-Thiadiazole Thione Moiety"

_molecules, 2024, doi:10.3390/molecules29020285_

Round 1

Reviewer 1 Report

Comments and Suggestions for Authors

The manuscript entitled “Design, Synthesis, Antibacterial, and Antifungal Evaluation of Phenylthiazole Derivatives Containing a 1,3,4-Thiadiazole Thione Moiety” (Manuscript ID:  Molecules-2807198 ) by Li et al. describes the design and synthesis of 1,3,4-Thiadiazole thione containing phenylthiazole derivatives and their evaluation for antibacterial and antifungal properties. Authors also studied the structure activity relationship of these molecules. However, the manuscript needs minor changes before accepted for publication.

1.     Provide the complete scientific name of the bacteria/fungi in the first instance in the manuscript. Italicise the scientific name in the entire manuscript.

2.     Line 38, Include suitable reference for ginger.

3.     Figure 1. Structure of Bismerthiazol needs attention.

4.     Section 2.1.

a.     Line 80, Describe step 1 in the text.

b.     Line 84, ring-closed to ring-cyclized

c.     Line 92, Did authors observe any CF3 splitting in 13C NMR spectrum of compound 5O.

d.     Line 107, Replace Figure with Scheme

e.     Figure 3 (Scheme 1) Compound Numbering needs attention. 2a missing, 4p twice etc. Reaction conditions: Temperatures, Time and Yields: All the details should be represented on the scheme.

5.     Table 1. Include details in the legend, number of times experiment is repeated, E1?, spacing errors need to be corrected.

6.     References need attention, formatting errors to be corrected (Ref 23, Journal name different font.

7.     Any comments on morphological, physicochemical changes induced by these molecules?

Comments on the Quality of English Language

Language polishing suggested

Reviewer 2 Report

Comments and Suggestions for Authors

A number of 1,3,4-thiadiazole derivatives were used as commercial pesticides and the 1,3,4-thiadiazole cycle is considered as a promising scaffold for agriculture. In the reviewed paper, 16 phenylthiazole derivatives containing a 1,3,4-thiadiazole thione moiety were designed and synthesized. The obtained results indicated that most of the synthesized compounds displayed antibacterial and antifungal activities. In particular, compound 5k (R= 3-OCF3) possessed the high antibacterial activities against R. solanacearum, which was superior to that of the commercial bactericide Thiodiazole copper. In addition, compound 5b (R=2-F) exhibited the potent antifungal activity against S. sclerotiorum, which was equivalent to that of the commercial fungicide Carbendazim. These products with fluorine substituents can be used as lead compounds in further research. The preliminary structure-activity relationship results were discussed. In my opinion, the obtained results are important and deserved to be published.  An important remark: if the obtained products are new compounds, then this should be noted in the text of the article, and in this case, the spectral data of new compounds have to be moved from Supporting Materials to the experimental part of the paper.

Comments on the Quality of English Language

Minor editing of English language required.
